# Daily Life Benefits and Usage Characteristics of Dynamic Arm Supports in Subjects with Neuromuscular Disorders

**DOI:** 10.3390/s20174864

**Published:** 2020-08-28

**Authors:** Johannes Essers, Alessio Murgia, Anneliek Peters, Kenneth Meijer

**Affiliations:** 1Department of Nutrition and Movement Sciences, NUTRIM School of Nutrition and Translational Research in Metabolism, Maastricht University, 6229ER Maastricht, The Netherlands; kenneth.meijer@maastrichtuniversity.nl; 2Department of Human Movement Sciences, University of Groningen, University Medical Center Groningen, 9713AV Groningen, The Netherlands; a.murgia@umcg.nl; 3Department of Rehabilitation Medicine, University of Groningen, University Medical Center Groningen, 9713AV Groningen, The Netherlands; a.a.peters@umcg.nl

**Keywords:** neuromuscular disorders, dynamic arm support, activity monitoring, motor performance, upper extremities

## Abstract

Neuromuscular disorders cause progressive muscular weakness, which limits upper extremity mobility and performance during activities of daily life. Dynamic arm supports can improve mobility and quality of life. However, their use is often discontinued over time for unclear reasons. This study aimed to evaluate whether users of dynamic arm supports demonstrate and perceive quantifiable mobility benefits over a period of two months. Nine users of dynamic arm supports were included in this observational study. They had different neuromuscular disorders and collectively used four different arm supports. They were observed for three consecutive weeks during which they were equipped with a multi-sensor network of accelerometers to assess the actual use of the arm support and they were asked to provide self-reports on the perceived benefits of the devices. Benefits were experienced mainly during anti-gravity activities and the measured use did not change over time. The self-reports provided contextual information in domains such as participation to social life, in addition to the sensor system. However self-reports overestimated the actual use by up to three-fold compared to the accelerometer measures. A combination of objective and subjective methods is recommended for meaningful and quantifiable mobility benefits during activities of daily life.

## 1. Introduction

Neuromuscular disorders affect 153 persons per 100,000 in the Netherlands and 160 persons per 100,000 worldwide [1,2]. One of the symptoms of neuromuscular disorders is muscular weakness, which is progressive in most cases and therefore increasingly limits upper extremity mobility and performance during activities of daily life (ADLs). Approximately 7–24% of individuals with a neuromuscular disorder use dynamic arm supports (DASs) [3,4], which provide gravity compensation and can improve mobility and quality of life [5,6,7,8]. Specifically, they facilitate limb motion against gravity [9,10,11,12,13,14], reduce efforts [8,12,13,14,15], and improve ADL performance [5,9,11,16]; thus, supporting the user’s overall activity and independence [10,15]. Studies have shown that the intended benefits of DASs are not completely realized [17,18], whereas most users seem satisfied with the DAS given to them, with continuous use reported up to 17 h per day [6]. However, over time most users no longer perceive these benefits and stop using the DAS altogether [17], which expectedly leads to a loss of function and reduced participation and quality of life. Experts believe that the disease progression makes it more difficult to operate the DAS; thus, contributing to the changed perception over time [18]. It is therefore important to understand the causes that lead to a DAS not being used any longer by first investigating the quality, or characteristics, of DAS usage.

Experts promote the integration of objective and subjective information on DAS usage [18,19], to cover different components of the International Classification of Functioning, Disability, and Health (ICF) model [20]. Objective information, such as improved mobility and reduced efforts, mostly reflects device effectiveness in the body function ICF component. Subjective information, such as user needs, wishes, and experiences, mostly cover the impact of a DAS in the activity and participation components of the ICF as well as environmental and personal factors. Ideally, these aspects should be combined and monitored over time to detect temporal changes in daily life behavior that would result in discontinuation.

Currently, monitoring methods in the field mainly rely on (subjective) self-reports, which are recommended to identify the reasons for DAS use [21]. Such methods are valuable to assess subjective factors, such as the perceived activity, benefits, and limitations, and possibly specific causes for discontinuation of use. However, they are also prone to bias, e.g., self-reported duration of DAS use was found to depend heavily on users’ expectations of, and reliance on, the device [6,9]. Similar bias was also present in studies evaluating the functional improvement where a patient’s perceived gain was higher than the gain detectable through clinical scales [22]. The low level of specificity and the bias in self-reports makes it difficult to distil the most important aspects of DAS usage [21]. Therefore, we need to move beyond self-reporting if we want to understand the benefits and limitations of a dynamic arm support for specific user groups.

Accelerometer-based activity monitoring overcomes these inadequacies in self-reporting by objectively quantifying daily life upper extremity activity [23,24,25,26,27]. This approach has been successfully applied in various populations such as children with neurodevelopmental disorders, stroke survivors, and upper limb prosthesis users. In previous work [24], a multi-sensor network was used to classify upper and lower arm activities of children with Duchenne Muscular Dystrophy during ADLs. The activity classifications, intensity, orientation, and frequency of arm elevations provided valuable insights into the daily activity levels, such as the timing, intensity, and duration of activities. Furthermore, the activity classifications correlated strongly with upper extremity functionality measured on a clinical scale (Brooke scale R: 0.73 ± 0.13), where less capable participants had lower activity levels and elevated their arms less frequently [24]. However, accelerometer-based activity monitoring has limited power to register and discriminate between postural ADLs, such as while holding a telephone to the ear or typing on a keyboard. These activities, which are also important indicators for DAS use, should therefore not be neglected [6] and may be better captured via self-reports. It is evident that accelerometers and self-reports provide more detailed insight in the reasons for discontinued use and guide DAS development. However, this combined approach has still not been used to understand the usage characteristics of dynamic arm supports in persons with neuromuscular disorders.

The aim of the current study is to determine whether DASs produce quantifiable upper and lower arm mobility benefits that impact specific ADLs. These benefits are derived from the duration, intensity, and frequency of reoccurring activities performed with and without a DAS. Furthermore, it will be investigated whether users also perceive these benefits, based on self-reporting assessments, and whether these benefits are consistent over time. An integrated activity-monitoring approach that exploits accelerometer sensor networks, in combination with self-reports, was adopted.

## 2. Materials and Methods

### 2.1. Participants

Potential participants were informed about this study through digital flyers advertisements within the networks of Dutch Association for Neuromuscular Diseases (Spierziekten Nederland, Baarn, The Netherlands), Focal Meditech (Tilburg, The Netherlands), Maastricht University Medical Center+ (Maastricht, The Netherlands), and University Medical Center Groningen (Groningen, The Netherlands). Interested participants were pre-screened and included when older than 18 years, used a DAS at home, had a diagnosed disease resulting in muscular weakness, did not have other conditions that limited upper extremity movement (i.e., tremors), and could give written informed consent. Three researchers visited the participants at home to acquire the informed consent, provide a diary, and place activity sensors. The central Medical Ethical Committee of Maastricht University Medical Center+ approved the study (17-4-031.1), which was carried out in accordance with the guidelines of the Helsinki protocol.

Nine participants (4M:5F, 51 ± 14 years) diagnosed with a neuromuscular disorder were included in this study (Table 1). The participants used one or two DAS devices of the same type. Seven participants were entirely wheelchair-bound with the DAS mounted on the wheelchair. One participant (P3) was ambulant and had a chair-mounted DAS in the kitchen area. Another participant (P5) was ambulant but required a walker and had a wheelchair-mounted DAS. Four participants (P3, 5, 8, and 9) were monitored during three measurement periods, two (P4, 7) during two periods, and three (P1, 2, and 6) during one period of seven consecutive days. The monitored side for P3 and P8 deviated from the dominant side because P3 mostly used his left-sided device and P8 did not have access to her right-sided device due to a scheduled maintenance.

Participants collectively owned four different types of DAS; the Armon Edero [28], Dowing [29], Gowing [30], and Sling [31]. The Armon Edero, Dowing, and Sling are passive support devices with adjustable gravity compensation. The Armon Edero and Dowing use adjustable springs, whereas the Sling uses counterweights to support the weight of the arm. The Gowing is a hybrid device that provides spring-actuated passive support with an addition of motorized actuators to adjust the springs and provide active support. These devices are relatively easy to put on and off, allowing the user to switch between use and non-use with little effort.

### 2.2. Activity Sensors

DAS use and upper extremity motions were monitored using activity sensors, Figure 1. The activity sensors (MOX, Maastricht Instruments, Maastricht, The Netherlands) were 3D accelerometers with inbuilt data loggers capable of at least seven days of recording with a sample rate of 25 Hz [32,33]. Sensors were placed similarly to previous work [24]; on the lateral side of the upper arm (UA), on the lower arm at the wrist (LA), and on the device’s base (DB), and in addition, on the supporting brace of the DAS in line with the wrist sensor (SB). The UA sensor provided information on the utilization of the shoulder joint, the LA sensor on the utilization of the shoulder and elbow joints combined, the LA and SB sensors combination on the use of the DAS, and the DB sensor on the transportation of the whole device. Participants were asked to wear the sensors continuously up to a maximum of 21 monitoring days divided into three periods of seven consecutive days with an interval of at least fourteen days between periods to monitor changes over time. Data analysis focused on a 24-h representation of activity. Therefore, days when sensors were worn for less than 24 h or when sensors had technical issues were excluded from the data analysis.

### 2.3. Data Processing

Recorded accelerations were processed to provide information on the daily intensity (counts/s), defined as an integrated vector [18] and body segment orientation based on the gravity vector (pitch: 0–180 degrees) over time [24] (Figure 2, Appendix A
Figure A1). This information was further transformed to express the DAS usage characteristics in terms of duration, frequency, and activity levels (Table 2).

First, the tri-axial accelerations were filtered with a fourth order Butterworth 0.025 to 7.5 Hz band pass filter to calculate the intensity for each sensor (Appendix A, Figure A2, Figure A3, Figure A4 and Figure A5). A threshold (1.125 counts/s) was determined during system calibrations to distinguish between still and motion based on the collective sensor noise level of a one-minute recording while sensors laid still and a one-minute recording of slow movements. Furthermore, we have verified the threshold with participant data and found the threshold to clearly distinguish between resting periods and activity bouts, also over longer periods of time. All intensity data were categorized per second as still or motion and corrected for non-stationary periods of the device’s base as still. The lower arm sensor determined the periods of activity (motion). Within active periods, the support brace sensor determined the periods of DAS non-use (still) and use (motion). The periods of activity, non-use, and use were expressed as cumulative minutes per day. Furthermore, the intensity levels of non-use and use were extracted for the upper and lower arm and period occurrences of non-use and use for the lower arm only. The intensity levels were expressed as cumulative counts per day and the period occurrences (episodes) counted per day. The intensity data processing resulted in the outcome parameters periods of activity, non-use, and use, and the intensity levels and episodes of non-use and use of the upper and lower arm (Figure 2).

Second, accelerations of the axis in line with gravity, in a neutral body position, were filtered with an eight order Butterworth 2 Hz low pass filter to calculate the orientation from the arccosine per body-worn sensor (Appendix A, Figure A2 and Figure A3). Orientation data were categorized for each sample (1/25 s) as low or high with a double threshold to filter threshold fluctuations. Samples had to be below the low threshold (UA: 40° and LA: 115°) or above the high threshold (UA: 50° and LA: 125°) to be classified as such and those between thresholds were placed in the same category as the precedent sample. The upper arm thresholds were chosen to differentiate between the low (<45°) and the middle and high orientations (>45°) of the upper arm, as used by van der Geest et al. 2019 [24], with a tolerance margin of ±5°. The lower arm thresholds represent the inclination, 120° with a tolerance margin of ±5°, where motions become more challenging with a DAS. This is because the application of the vertical force, normally on the entire lower arm, is being positioned towards the elbow with greater inclination and users therefore receive less support for elbow flexion. These thresholds were then verified during system calibrations for several ADLs; simulated eating/drinking, reaching above shoulder level, and typing on a keyboard, at a slow, normal, and fast pace without and with a DAS. Time spent in high orientations were added for the upper and lower arm respectively and expressed as minutes per day. Furthermore, transfers from a low to high state were counted to represent the respective arm elevation frequency expressed as occurrences per day. The orientation data processing resulted in the outcome parameters time in a high orientation and arm elevation frequency, both during non-use and use of the upper and lower arm (Figure 2).

Finally, the parameters related to intensity levels, episodes, time in high orientation, and arm elevation were also normalized for DAS use and expressed as percentages of non-use and use, respectively (Table 2).

### 2.4. Self-Reports

Diaries were used to extract self-reported ADLs that are considered reoccurring within a day or week, such as eating/drinking, self-care, and work. Furthermore, the participant was asked retrospectively to answer five questions per monitoring period concerning DAS benefits and limitations. The questions were “describe motions or activities that (1) were unsuccessful without DAS, (2) were unsuccessful with DAS, (3) required increased effort with DAS, (4) were only possible without DAS, and (5) were only possible with DAS”. In addition, we retrospectively inquired about the participants’ (1) perceived daily activity, (2) perceived daily use, (3) perceived benefit from the DAS on a 0 to 100 scale (0 = “I don’t use it at all” and 100 = “I use it continuously”), and (4) the ratio between left and right arm involvement in their activities on a 0 to 100 scale. Answers were collected after initial data quality analysis, two to four weeks after all monitoring periods, to minimize the influence on participants’ awareness of DAS use and ADL performance.

### 2.5. Data Synthesis

Monitored daily activity levels were divided in primary and secondary outcome parameters (without and with DAS) based on the data processing sequence. Primary outcome parameters were quantified as the time spend active and secondary as (1) the time spend with the arm elevated, (2) the frequency of activities episodes and (3) arm elevation, and (4) the intensity levels of the activities. Device benefits were quantified as the effect sizes of secondary outcome parameters during DAS use compared with non-use. The perceived averaged daily use, collected once, was tested for significant difference with the monitored daily use (primary outcome parameter). The daily collected perceived and monitored device benefits were compared on the similarity of self-reported activity benefits and the effect sizes of daily activity levels (secondary outcome parameters).

### 2.6. Statistical Analysis

The perceived daily DAS use was compared to the averaged monitored equivalent using a paired sample Wilcoxon signed rank test (SPSS) [34]. Changes in monitored activity and DAS use over the monitoring periods were investigated with a non-paired sample Mann–Whitney U test of period combinations 1–2, 2–3, and 1–3 within each subject where possible. Alpha levels were set at 0.025.

Cohen’s d effect sizes were calculated within subjects for the secondary outcome parameters (absolute and normalized) using the formula:(1)$$d\;=\frac{\left(m1-m2\right)}{\sqrt{s1^{2}+s2^{2}-\left(2*r*s1*s2\right)}}$$

Where m1 and m2 represent the means during use and non-use respectively, and the s1 and s2 the standard deviations during use and non-use, respectively. Pearson’s correlation coefficient (r) was calculated between use and non-use over the processed days. The effect sizes were calculated for each participant and as a group for the secondary outcome parameters, time spend in high, intensity levels, and elevations for the upper and lower arm sensors, and for episodes of the lower arm sensor. Effect sizes were ranged as small: 0.20–0.50, medium: 0.50–0.80, and large: >0.80 [35]. Parameters with a group effect size of medium and above (>0.50) were considered a mobility benefit resulting from DAS use. Changes in effect sizes over the periods were investigated with a paired sample Wilcoxon signed-rank test of period combinations 1–2, 2–3, and 1–3, where possible. Alpha levels were set at 0.025.

## 3. Results

The benefit of DAS use was quantified by comparing use and non-use periods of secondary daily activity levels. Effect size of DAS use was medium to large for normalized elevations of the upper (Cohen’s d: 0.6, *n* = 3) and lower arm (Cohen’s d: 1.0, *n* = 4), and large for normalized episodes of the lower arm (Cohen’s d: 1.7, *n* = 8) (Appendix A, Table A1). Other normalized parameters did not show a medium or above group effect and absolute values showed negative effects. The effect sizes over the three periods were not significantly different (Appendix A, Figure A6, Figure A7, Figure A8 and Figure A9). Daily activity levels were reported in the appendix (Appendix A, Figure A10, Figure A11, Figure A12 and Figure A13).

Reoccurring activities were mostly related to self-care (eating/drinking, hygiene, and house choirs) and computer activities (Table 3). The consensus from self-reports was that the DAS facilitated these activities and those involving reaching above shoulder level. Activities that involved forearm rotation, wrist motion, or large motions were mostly limited with a DAS. The averaged DAS benefits were rated as 83 ± 15% (Appendix A, Table A2). Furthermore, participants indicated that their monitored limb was used 60 ± 10% for their daily activities.

Participants perceived their averaged daily DAS use as more than monitored (*p*: 0.015) (Figure 3). Accelerometer data indicated that participants were on average active for roughly 561 ± 149 min a day of which 94 ± 77 min, or 18 ± 15%, was with a DAS (Figure 3). In contrast, participants reported an average of 430 ± 280 min being active a day of which 283 ± 212 min, or 74 ± 31%, with a DAS (Appendix A, Table A2). Furthermore, participants showed consistent activity and DAS use over time, except for P8 where the first period was lower (*p* < 0.010) compared to the latter two (Figure 4). The participant had stated in the diary to be ill and not very active in that period.

## 4. Discussion

This is, to the best of our knowledge, the first study that included a multi-sensor network of accelerometers in combination with self-reported activity to monitor the usage characteristics of dynamic arm supports (four different types) in people with neuromuscular disorders in a home environment. The primary results were that (1) the DASs facilitated motions against gravity and enhanced the occurrence of these activities and (2) that in our study population the objectively measured use of the DAS did not change over a two-month period. Furthermore, it was found that self-reports seriously (3-fold) overestimated the time spent using the arm support. The self-reports did yield more detailed information on the circumstances when the arm support was beneficial such as prolonged computer work, house choirs, and personal hygiene. In addition, it gave detailed information on the conditions in which the arm support was considered to limit activities, such as in activities involving wrist movements, forearm rotation, and large motions of the arm. This study clearly shows the benefits of a combined approach in quantifying the benefits and limitations of dynamic arm supports for activities in daily life, although several aspects need to be considered to optimize the approach.

The integration of accelerometers with self-report measures in a home environment provided a quantifiable comparison of mobility benefits to describe facilitated ADLs. As expected, repetitive motions, especially against gravity, were facilitated by DASs and perceived as beneficial in reoccurring ADLs, eating/drinking and touching their head. However, several perceived ADL benefits or limitations were not identifiable from accelerometer data because the context was too general (house choirs and personal hygiene) or they concerned aspects which were not measured (wrist movements and device range of motion). Furthermore, the heterogeneous effect sizes across participants and parameters favors the focus on participant-specific rather than general mobility benefits. Therefore, device effectiveness would be best determined on an individual basis. This should be organized according with what a person can do (motor capability), what a person wants to do (needs and wishes), and what a person does in daily life (motor performance) [18]. However, common benefits, such as activity frequency and arm elevation, could provide initial indications of device effectiveness for essential activities, such as eating/drinking [15,36].

The ability to monitor changes in the use of the dynamic arm support over time could potentially help clinicians and developers optimize the system for the user [18,19]. The two-month monitoring period used in this study was too short to capture discontinuation or even reveal large changes in usage, although one participant showed a large increase in activity from week 1 to weeks 2 and 3, which via the self-reports could be traced to a recovery from illness. Gradual changes in use could therefore provide first indications on disease progression, depending on the expected progression rate. The monitoring period should therefore be aligned with the expected progression rate, which for some neuromuscular disorders could expand to several months or years [37]. Essentially, a DAS should facilitate the use of muscles and independence in activities of daily life as important aspects of health, physically and mentally. Therefore, a better device match is preferred, which depends on the user’s changing capabilities, needs, and wishes. DAS developers could use these longitudinal assessments to optimize device benefits and clinicians would be able to alter therapy or DAS type in individual cases so as to limit future discontinuation [18].

However, future research should reduce the physical and mental burden by minimizing the required number of worn sensors and diary input [24]. It was found that for some participants wearing two sensors and keeping a diary for seven consecutive days was a heavy burden, both physically as mentally. Solutions that use a single sensor at the wrist and use experience sampling of the satisfaction via an app, might overcome these issues an offer a feasible and participant-friendly option to obtain this valuable information.

Studies on whole body activity monitoring show that participants tend to overestimate their activity levels and underestimate their sedentary/resting episodes [38,39,40]. As expected, participants in the current study also mostly overestimated their use compared to those monitored with the multi-sensor network. However, the self-reported daily activity levels, mobility benefits, and satisfaction were comparable to those in other self-reported studies [6,8,17]. The monitored data showed that participants in this study spent about as much time being physically active as unimpaired people [23]. Furthermore, they lifted their arms about as often (28.9 ± 15.4 per hour) as children with Duchenne [24]. It should be realized that objective and subjective outcomes relate to different ICF domains. The multi-sensor network measures mostly the body functionality, while self-reports naturally tend to cover activity and participation [15,17,18]. Even though the ICF model components interact with each other, the measurements methods may not reflect the same aspects of DAS usage characteristics. To clarify, DAS users may have utilized the device for postural support or might have had breaks during activities that were experienced as a continuous activity but not monitored as such by the multi-sensor network. For example, arm motion is not required when typing and reading during computer work or chewing during eating, but these activities can still be considered as using the DAS. This could explain the differences in DAS use to some extent although not the observed daily difference of 3 h. Furthermore, the averaged monitored use is considerably low (<20%) and varies greatly between and within participants. It is therefore unclear whether self-reports are sufficient to reflect the daily use or if postural activities present such a large aspect of DAS use. However, this would have a limited effect for within-participant comparisons, which are more important for future directions, such as determining individual device effectiveness and performing longitudinal measurements [17,18,19].

As expected, larger effect sizes were generally more prominent in the lower arm than the upper arm, as the lower arm is primarily supported and motion in the upper arm influences the first. Furthermore, effect sizes varied greatly between participants which could reflect environmental and personal factors, such as device-specific facilitations or otherwise undoable ADLs and disease-specific affected muscle regions. In muscular dystrophy, such as Duchenne and limb girdle, proximal regions of the upper extremity are affected first which limits upper arm elevations [4,41]. In addition, ADLs such as eating and drinking can be performed with various contributions from the shoulder and elbow joint and may not depend on the ability to utilize both. Therefore, investigating the upper extremity as separate segments might reveal benefits corresponding to specific environmental and personal factors. However, the upper arm sensor would be redundant when investigating the collective mobility benefits reported in this study.

This study provided several significant contributions to a better understanding of DAS usage characteristics. First, the complementary information from objective and subjective measures provided better insights in DAS use and benefits, specifically about overestimation in self-reporting and sensitivity to energy loss. Second, mobility benefits were up to now not yet quantified in a home environment. Consequently, this study highlighted common benefits and the need for individual assessment. Third, multiple assessments up to a two-month period provided first-time evidence of actual use and revealed longitudinal consistency of daily activities. However, there were some limitations present in this study. First, the majority of participants did not complete the proposed 21 days, due to the imposed physical and mental burden (*n* = 5) or technical issues (*n* = 3), due to a limited battery life and noise (data clipping and repetitive single sample acceleration peaks). Sensor and diary reductions might lower the burden and thus allow a larger inclusion of participants and recordings. Second, the current study reports the mobility benefits after the DAS has been integrated in daily life and might not fully reflect the intended life-changing benefits. Furthermore, participant-specific benefits were not investigated, as residual arm movement and disease progression were not the target of this study. However, the current method could be used to monitor the DAS usage characteristics pre and post DAS integration in daily life to investigate the realization of intended benefits for several purposes such as device development and user-device optimization [18].

## 5. Conclusions

This study showed that the integration of a multi-sensor network and self-reports gives additive information on the use and benefits of dynamic arm supports. It has been shown that movements performed at home were executed more frequently with an arm support and that participants benefitted from the devices in important tasks such as eating and drinking and touching their head. Further simplification and integration of the assessments to address the relevant ICF domains is necessary to translate users’ needs and wishes to mobility benefits and determine device effectiveness.

## Figures and Tables

**Figure 1 sensors-20-04864-f001:**
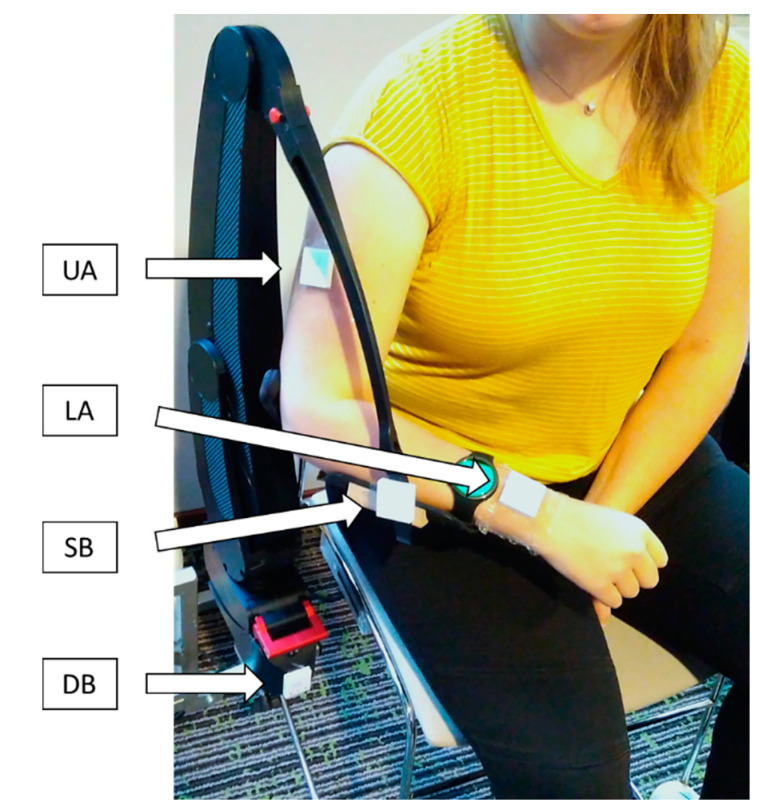
Measurement setup demonstration on a healthy volunteer. UA: Upper Arm, LA: Lower Arm, SB: Support Brace, and DB: Device Base.

**Figure 2 sensors-20-04864-f002:**
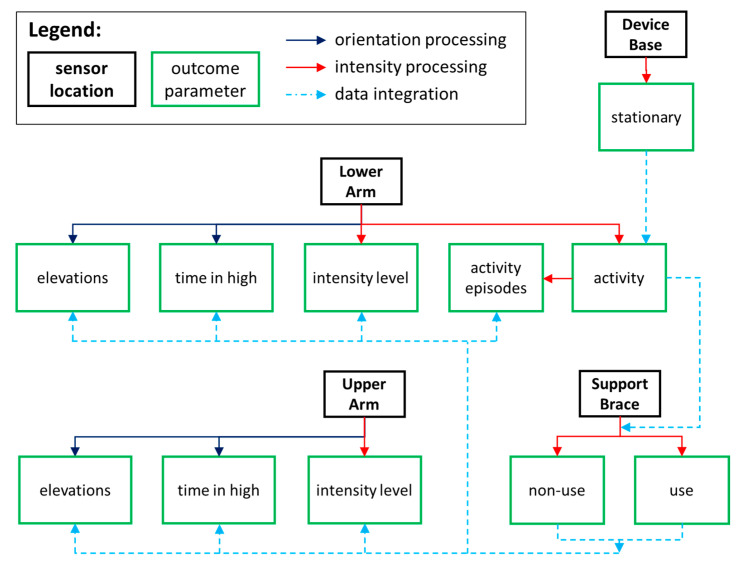
Multi-sensor network processing for orientation and intensity related parameters and integration of data between sensors.

**Figure 3 sensors-20-04864-f003:**
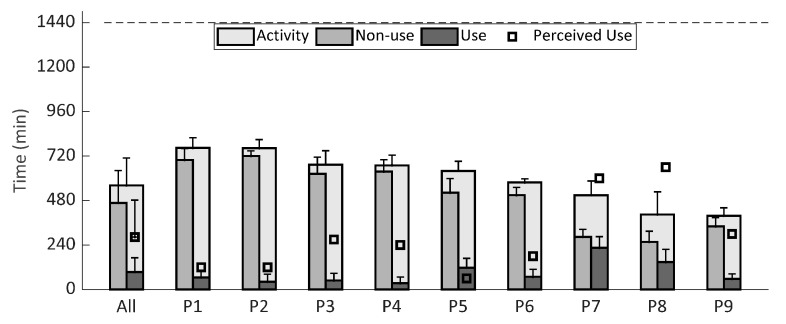
Averaged monitored daily activity, non-use, and use expressed in minutes as bars and perceived use as squares, both with one standard deviation. The dotted line represents a complete day. Participants were sorted on activity time in a descending order for visualization purposes.

**Figure 4 sensors-20-04864-f004:**
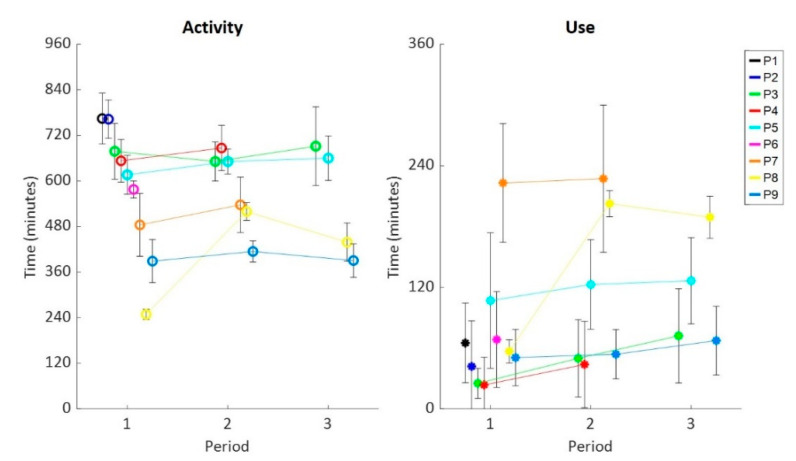
Activity (left; o) and use (right; *) per individual (colors) over the three periods. Participants were sorted on averaged activity time of all periods in a descending order for visualization purposes.

**Table 1 sensors-20-04864-t001:** Participant characteristics. BL: Bilateral.

	Gender (M/F)	Age (Y)	Diagnosis	Dynamic Arm Support	Dominant Side	Monitored Side	Total Days
P1	F	61	Congenital Myopathy	Gowing	Equal	Left	3
P2	M	44	Amyotrophic Lateral Sclerosis	Gowing	Right	Right	6
P3	M	58	Progressive Spinal Muscular Atrophy	Dowing (BL)	Right	Left	21
P4	F	30	Limb Girdle Muscular Dystrophy	Armon Edero	Right	Right	12
P5	F	63	Desminopathy	Sling (BL)	Right	Right	16
P6	F	60	Limb Girdle Muscular Dystrophy	Dowing	Right	Right	4
P7	M	55	Limb Girdle Muscular Dystrophy	Gowing	Right	Right	13
P8	F	60	Spinal Muscular Atrophy type 3	Gowing (BL)	Right	Left	17
P9	M	27	Duchenne Muscular Dystrophy	Gowing (BL)	Right	Right	21

**Table 2 sensors-20-04864-t002:** Overview of monitored outcome parameters that represent dynamic arm supports (DASs) usage characteristics.

Primary	Secondary		
**Time** absolute (min/day)	**Time** absolute (min/day) normalized (% activity)	**Activity levels** absolute (counts/day) normalized (counts/min)	**Frequency** absolute (#/day) normalized (#/min)
activity use non-use	UA use in high LA use in high UA non-use in high LA non-use in high	UA use intensity LA use intensity UA non-use intensity LA non-use intensity	UA use elevations LA use elevations UA non-use elevations LA non-use elevations LA use episodes LA non-use episodes

**Table 3 sensors-20-04864-t003:** Summary of the self-reported reoccurring, facilitated, and limiting activities.

Reoccurring Activities	Only Possible with Device	Only Possible without Device
Eating/drinking (P1–5,7–9) Personal hygiene (P1–5,7–9) Computer activities (P1–5,7–9) House choirs and cooking (P1,4,5,7,8) Touch head (P7,8,9)	Eating/drinking (P4,5,8,9) Extended time computer work (P5,8) Touching head (P7,8) Personal hygiene (P8)	Driving a car (P4) Motions beyond the ambulant chair (P5) Typing (P7)
	Unsuccessful with device	Unsuccessful without device
	Proper pronation/supination (P3) Maintaining arm within DAS (P3,4) Reaching below waist level (P3–5) Personal hygiene (P4) Folding laundry (P5) Support of wrist during hand to mouth (P5) Opening door handle (P7) Washing hand (P9)	Personal hygiene (P2) Reaching above shoulder (P3,4,5,7) House choirs/cooking (P4) Shake hands (P5) Flush toilet (P5) Opening door handle (P7) Eating/drinking (P7) Almost everything (P9)

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
