# Peer review of "Daily Life Benefits and Usage Characteristics of Dynamic Arm Supports in Subjects with Neuromuscular Disorders"

_sensors, 2020, doi:10.3390/s20174864_

Round 1

Reviewer 1 Report

In this study inertial sensors are used to evaluate the benefit of using a dynamic arm support in subjects affected by neuromuscular disorders. The multi-sensors network of accelerometers used here has been already presented in a previous work from some of the authors. Here, it was used to monitor 9 subjects with different mobility problems for up to 21 days over a period of around 2 months.

Current monitoring methods mostly rely on self-reports, which can be heavily subjected to individual bias. More systematic and objective monitoring methods are therefore needed to complement the conventional self-reports. The reasoning behind this study is well explained by the authors.

I do not have a medical background so my judge is very limited on that prospective, I reviewed the article from a technical and a scientific point of view, with my interest set mainly on the strictness of the scientific method. To me, the idea of monitoring these particular population of patients with a inertial sensors network makes completely sense. I liked the article and I appreciated the efforts behind it. However, I found few issue that should be clarified.

To me, it is unclear how and if the proposed methods can prevent motions in only one part of the arm to be propagated and measured as active motions in other parts or the arm. If, for example, a participant had residual movement mainly in the UA but not much in the LA, how were the movements of the UA registered from the sensor in the LA? Would they be counted as functional movements of the LA too? How would these UA movements reflect into the intensity, duration and frequency outcomes of LA? Is this relevant for the approach proposed, or is it the aim to track any motion in the patient, regardless to its functionality? To my understanding of the Introduction, it sounds as it relevant to quantify mobility benefits for upper and lower arm separately. In such case, it is unclear which are the limitations of this approach. I would appreciate some clarifications on this matter but I also suggest to argument this aspect somewhere in the manuscript.

In the beginning of page4, "technical issues" are mentioned. Knowing the nature of these technical issues is of relevance for any reader to understand how this approach might be close (or far) from a longer clinical implementation. More details must be added.

In the Introduction, a big focus is set on the importance of long-term objective monitoring of the use of DASs to understand why, even though such devices are perceived as useful, patients still tend to stop using them: "Yet, over time most users no longer perceive these benefits and stop using the DAS altogether, which expectedly leads to a loss of function and reduced participation and quality of life". Part of this is confirmed in this study, where it was found a clear trend of the participants to overestimate the perceived use of the DAS, showing a strong bias towards this technology. However, the 2 months duration of this study does not seem sufficient to see participants abandoning their device, suggesting the need for a longer test period and more test subjects. This prospective must be discussed in the manuscript. Moreover, it is unclear to me what is the official position of this study in regard to the issue of patients stopping to use the DASs. I believe that a position should be taken, or at least such matter must be treated in the Discussion section.

The study does not mention nor treat the case of tremors at all. However, I believe that unintentional tremors might occur in patients with the neuromuscular disorders as the ones treated in this study. Was this an issue with your participants? Was it considered in your selection criteria? If so, it must be clearly stated. Then, how would your inertial sensors based approach work in case of patients with tremors? This aspect should be mentioned in the manuscript.

How was the 1.125 counts/second time threshold verified for reliability? Was a unique calibration sufficient for all participants? Were re-calibration of the device needed over time? It is a crucial parameter for the monitoring performed in this study so it deserves more attention in the manuscript, please add info. The same indication goes for the all other monitoring parameters like the threshold angles.

Figure A6, A7, A8, A9 miss a legend for the identification of the participants (as in Figure 4). A reader must be able to easily understand ALL data presented so to make his/her own interpretations possible.

On the same idea of allowing readers to make their own analysis interpretation, I think that table 1 should include also a brief description of the residual arm movements at the beginning and at the end of the monitoring time for each participant. At least, if any difference was found in any of the participants, this must be stated in the manuscript.

Even though this information is correctly included in the captions, I also suggest to add a legend for "use" and "non-use" data in the Figures A10, A11, A12 and A13.

I have no comments in regard to the language, the manuscript is well written and easy to understand. I have only few minor suggestions in the following.

Page3, line98: Adjust the following sentence "Nine participants were included in this study and were diagnosed with a pathology involving muscular weakness.." into something like "Nine subjects diagnosed with NMD were involved in this study.."

Page3, line107: All four devices should be named at first, and then presented in details. Otherwise it is confusing for the reader.

Page3, line123: I suggest to refer always to "monitoring days" rather than "measuments days".

Page6, line216: Please avoid the use of informal forms like "very large" or "huge".

Page9, line320: Please re-write the following sentence as it is a critical sentence in the manuscript but, at the moment, it is very confusing for the reader. "It was shown that due to the support motions at home, especially against gravity, were performed more frequently and participants benefited from the devices in vital tasks such as eating and drinking and touching their head.

I believe the NMD acronym can be avoided (it is used only twice).

Reviewer 2 Report

This paper  presents an integrated activity-monitoring approach that
 exploits accelerometer sensor networks, in combination with self-reports to check the benefit of a proposed procedure.

The procedure and the reason is very well explained in the paper togheter with the approach the methodology ad the discussion.

To strenghten the soundness of the paper i will reference this paper that characterized the motion of the upper limb using people of different ethnicity using similar sensor network:

Cafolla, D., Ceccarelli, M. An experimental validation of a novel humanoid torso (2017) Robotics and Autonomous Systems, 91, pp. 299-313.

After this very minor revision the paper can be accepted in my opinion since it is appropriate for the issue

Reviewer 3 Report

This paper reported a procedure to monitor the DAS usage with a sensor network and self-report. The results validate the results that patients tend to report more DAS usage than what they real does. 

I have two major consents about this paper:

1) what is the significant contribution of this work? To me, it stops at validating previous reported results and reporting some knowledge people already know. If that is the case, the readers do not get new things from it.

2) the sensor network should provide the information of both upper arm and lower arm. But the authors does not explore what this additional information will bring. If the activity of the arm is the only thing, which people wants to monitor, a 3D accelerator on the wrist should get most of the motions on someone with muscle weakness.  

Round 2

Reviewer 1 Report

Thanks for following all my suggestions, I appreciate the reviewing efforts. I believe the manuscript is clearer now, and ready for publication.

Author Response

We thank the reviewer for the previous suggestions and were happy to improve the manuscript.

Reviewer 3 Report

The reviewer does not have further comments on the paper, but still want to clarify:

one of the major selling point is the body area network, which evaluates both upper arm and lower arm. However, this novel capacity of the network generates very limited results, which are reported in the paper (only the D-values are briefly mentioned and everything else is in appendix). I would recommend that additional results from the network should be included in the main section to support authors claims that monitoring both upper arms and lower arms is beneficial.
